# Neural circuit mechanisms of hierarchical sequence learning tested on large-scale recording data

**Toshitake Asabuki** [ID] [¤] *, **Prajakta Kokate** [ID], **Tomoki Fukai** [ID]

Neural Coding and Brain Computing Unit, Okinawa Institute of Science and Technology, Onna-son, Okinawa, Japan

¤ Current address: Bioengineering Department, Imperial College London, London, United Kingdom
* toshitake.asabuki@gmail.com

**Data Availability Statement:** All numerical datasets necessary to replicate the results shown in this article can easily be generated by numerical simulations with the software code provided below. No datasets were generated during this study. All

## Abstract

The brain performs various cognitive functions by learning the spatiotemporal salient features of the environment. This learning requires unsupervised segmentation of hierarchically organized spike sequences, but the underlying neural mechanism is only poorly understood. Here, we show that a recurrent gated network of neurons with dendrites can efficiently solve difficult segmentation tasks. In this model, multiplicative recurrent connections learn a context-dependent gating of dendro-somatic information transfers to minimize error in the prediction of somatic responses by the dendrites. Consequently, these connections filter the redundant input features represented by the dendrites but unnecessary in the given context. The model was tested on both synthetic and real neural data. In particular, the model was successful for segmenting multiple cell assemblies repeating in large-scale calcium imaging data containing thousands of cortical neurons. Our results suggest that recurrent gating of dendro-somatic signal transfers is crucial for cortical learning of context-dependent segmentation tasks.

## Author summary

The brain learns about the environment from continuous streams of information to generate adequate behavior. This is not easy when sensory and motor sequences are hierarchically organized. Some cortical regions jointly represent multiple levels of sequence hierarchy, but how local cortical circuits learn hierarchical sequences remains largely unknown. Evidence shows that the dendrites of cortical neurons learn redundant representations of sensory information compared to the soma, suggesting a filtering process within a neuron. Our model proposes that recurrent synaptic inputs multiplicatively regulate this intracellular process by gating dendrite-to-soma information transfers depending on the context of sequence learning. Furthermore, our model provides a powerful tool to analyze the spatiotemporal patterns of neural activity in large-scale recording data.

codes were written in Python3 with numpy 1.17.3 and scipy 0.18.1. Example program codes used for the present numerical simulations and data analysis are available at https://github.com/ToshitakeAsabuki/dendritic_gating.

**Funding:** This work was partly supported by KAKENHI (nos. 19H04994 and 18H05213) to T.F. T.A was supported by the SRS Research Assistantship of OIST. The funders had no role in study design, data collection and analysis, decision to publish, or preparation of the manuscript.

**Competing interests:** The authors have declared that no competing interests exist.

## Introduction

The ability of the brain to learn hierarchically organized sequences is fundamental to various cognitive functions such as language acquisition, motor skill learning, and memory processing [1–8]. To adequately process the cognitive implications of sequences, the brain has to generate context-dependent representations of sequence information. For instance, in language processing the brain may recognize "nueron" as a misspelling of "neuron" if the brain knows the word "neuron" but not the word "nueron". However, the brain recognizes "affect" and "effect" as different words even if the two words are very similar. "Break" and "brake" are also different words although these words combine the same letters in different serial orders. The brain can also recognize the same word presented in different temporal lengths. All these examples suggest the inherent flexibility of context-dependent sequence learning in the brain. However, the neural mechanisms underlying this flexible learning, which occurs in an unsupervised manner, remain elusive.

Segmentation or chunking of sensory and motor information is at the core of the context-dependent analysis of hierarchically organized sequences [9–12]. However, little is known about the neural representations and learning mechanisms of hierarchical sequences. Recently, neurons encoding long-range temporal correlations in the song structure were found in the higher vocal center of songbirds [13]. These neurons responded differently to the same syllables (the basic elements of bird song) depending on the preceding phrases (constituted by several syllables) or the succeeding phrases. Different responses of the same neurons in different sequential contexts were also found in the monkey supplementary motor area [14,15]. Hierarchical sequences are often assumed to mirror the hierarchical organization of brain regions. However, the human premotor cortex jointly represents movement chunks and their sequences [16] and linguistic processing in humans also lacks an orderly anatomical representation of sequential context [17].

Unsupervised, context-dependent segmentation is difficult in computational models. Recurrent network models can generate rich sequential dynamics, but these networks are typically trained by a supervised method. Spike-timing-dependent plasticity was used for unsupervised segmentation of input sequences in a recurrent network model [18]. However, while the model worked for simple hierarchical spike sequences, it could not learn context-dependent representations for overlapping spike sequences. Single-cell computation with dendrites could solve a variety of temporal feature analysis including the unsupervised segmentation of hierarchical spike sequences [19], supporting the role of dendrites in sequence processing [20]. However, context-dependent segmentation was also difficult for this model. Although the segmentation problem has been partially solved, recurrent connections alone or dendrites alone are insufficient for solving the difficult segmentation tasks such as exemplified in the beginning of this article.

Here, we demonstrate that a combination of dendritic computation and a recurrent gating dramatically improves the ability of neural networks to context-dependently segment hierarchical spike sequences. Our central hypothesis is that recurrent synaptic input multiplicatively regulates the degree of gating of instantaneous current flows from the dendrites to the soma. We derive an optimal learning rule for afferent and recurrent synapses to minimize a prediction error. The resultant dendrites generally learn redundant representation of multiple sequence elements while recurrent input learns to selectively gate the dendritic activity suitable for the given context. Recurrent networks with gating synapses were recently used in supervised sequence learning [21].

There is an increasing need for efficient methods to detect and analyze the characteristic spatiotemporal patterns of activity in large-scale neural recording data. We demonstrate that

the proposed model can efficiently detect cell-assembly structures in large-scale calcium imaging data. We show two example cases of such analysis in the hippocampus and visual cortex of behaving rodents. In particular, the latter dataset contains the activity of tremendously many neurons (~ 6,500), and analyzing the fine-scale spatiotemporal structure of activity patterns is computationally costly and difficult for any other methods. In contrast, the data size hardly affected the performance and speed of learning in our model. Surprisingly, the efficiency was even somewhat higher for larger data sizes. These results highlighted the crucial role of recurrent gating in amplifying the weak signature of cell assembly structure detected by the dendrites.

## Results

### The dendritic computation with recurrent gating network

Our recurrent network model consists of two-compartment neurons with somatic and dendritic compartments (Fig 1A and 1B). The dendritic components receive hierarchically structured afferent input and recurrent synaptic input, and the sum of these inputs drives the activity of the dendritic component. Afferent and recurrent connections onto the dendritic component are plastic and can be either excitatory or inhibitory. Before training, the weights of these connections are initialized by a Gaussian distribution with mean zero (see Methods). The somatic component receives a nonmodifiable uniform feedback inhibition, which induces competition among neurons. A similar two-compartment model without recurrent inputs has been studied in segmentation problems [19]. Here, we hypothesize that recurrent synaptic input multiplicatively amplifies or attenuates a current flow from the dendrite to the soma in an input-dependent fashion: The stronger the recurrent input, the larger the dendro-somatic current flow. This "gating" effect is described by a non-linear function of recurrent input to the neuron and controls the instantaneous impact of dendritic activity on the soma (Fig 1C). As in the previous models [19,22], all synaptic weights were trained to minimize the prediction error between two compartments. However, we considered the gated rather than raw dendritic activity in the error term. The rule derived for afferent synapses is the same as our previous rule [19] except for the gated dendritic activity in the error term. The learning rule for gating recurrent connections is novel and depends on the raw dendritic activity prior to gating. We will show below that the recurrent-driven gating plays a pivotal role in the learning of flexible segmentation. Unless otherwise stated, below the results are shown for network models with multiplicative recurrent inputs but no additive ones. In this setting, afferent inputs can evoke large somatic responses if and only if both dendritic activity and gating effect are sufficiently strong. A network model with both additive and multiplicative recurrent inputs will be considered later.

### The role of recurrent-driven gating in complex segmentation tasks

We first demonstrate the segmentation of two spike pattern sequences (chunks) repeated in input spike trains (Fig 2A). Each chunk was a combination of three fixed spike patterns out of the total five: "A", "B", "C", "D" and "E", where the component pattern "E" appeared in both chunks. Therefore, the two chunks were mutually overlapped. Throughout this paper, we fixed the average firing rate of each input neuron at 5 Hz over the entire period of simulations, and chunks were separated by random spike trains with variable lengths of 50 to 400 ms. As training proceeded, the correlation between the somatic response and gated dendritic activity increased (S1 Fig). Interestingly, the trained network generated two distinct cell assemblies, each of which selectively responded to one of the chunks (Fig 2B and 2C). Notably, each cell assembly responded to the pattern "E" in a preferred chunk of the cell assembly but not in a non-preferred chunk (Fig 2C and 2D). A network with a constant gating function trained on

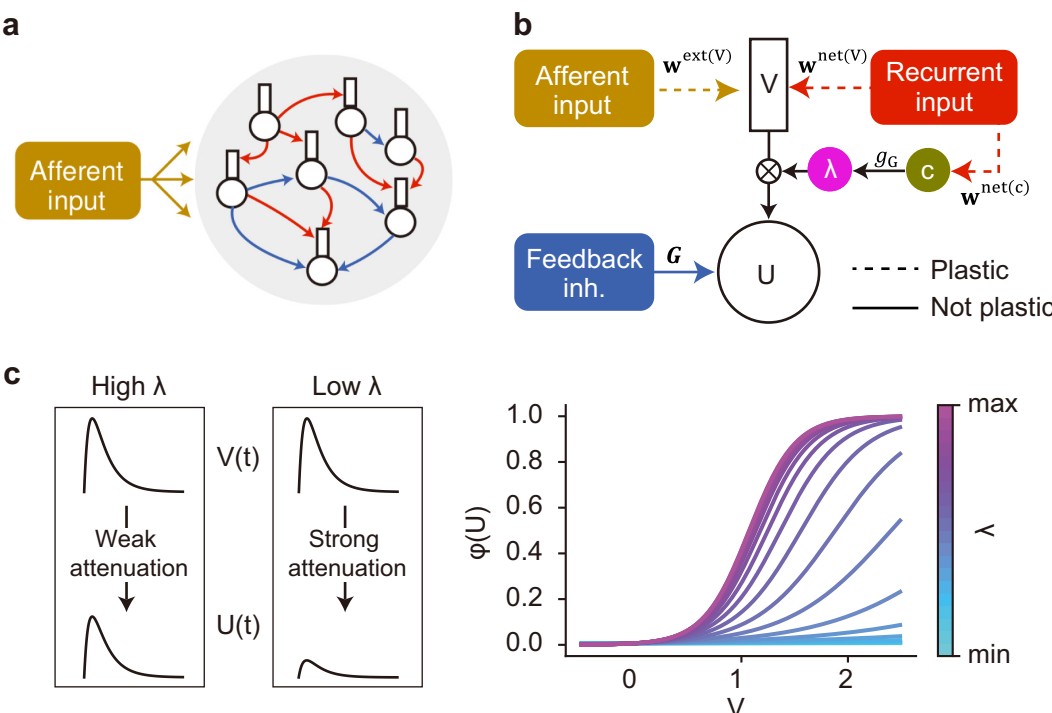

**Fig 1. A recurrent gated network of compartmentalized neurons.** (a) A network of randomly connected compartmentalized neurons is considered. The red and blue arrows indicate recurrent connections to the dendrite, and the feedback inhibition to the soma, respectively. (b) Each neuron model consists of a somatic compartment and a dendritic compartment. Dashed and solid arrows indicate plastic and non-plastic synaptic connections, respectively. Unless otherwise specified, the dendrite only receives afferent inputs. The somatic compartment integrates the dendritic activity and inhibition from other neurons. Gating factor λ is determined by recurrent inputs and regulates the instantaneous fraction of dendritic activity propagated to the soma. (c) The effect of gating factor on the somatic response is schematically illustrated (left). The relationship between the dendritic potential and somatic firing rate is shown for various values of gating factor (right).

the same afferent input failed to discriminate the pattern "E" in different chunks and consequently could not learn the chunks (S2 Fig). The result suggests the crucial role of the recurrent-driven gating in the segmentation task.

The above results suggest that this model can discriminate the context of sequences (i.e., the relationship between "E" and other component patterns in a chunk). However, it is also possible that the model separated the overlapping chunks merely relying on the component patterns that were not common between the two chunks and/or on the nonhomogeneous occurrence probabilities among the component patterns. To exclude these possibilities, we examined the case where different chunks shared all component patterns with equal frequencies (Fig 3A, left). In other words, the same component patterns occurred in different chunks in different orders (i.e., "ABCD", "DCBA", "BDAC"). During learning, the model was exposed to irregular spike trains recurring the three chunks intermittently (Fig 3A, right). The model developed distinct cell assemblies responding selectively to one of these chunks (Fig 3B and see S3A and S3B Fig), thus successfully discriminating the same components belonging to different chunks.

## The network mechanism of context-dependent gating

In this model, the recurrent-driven gating provides context-dependent signals necessary for segmenting overlapping chunks. To gain an insight into the role of recurrent gating in

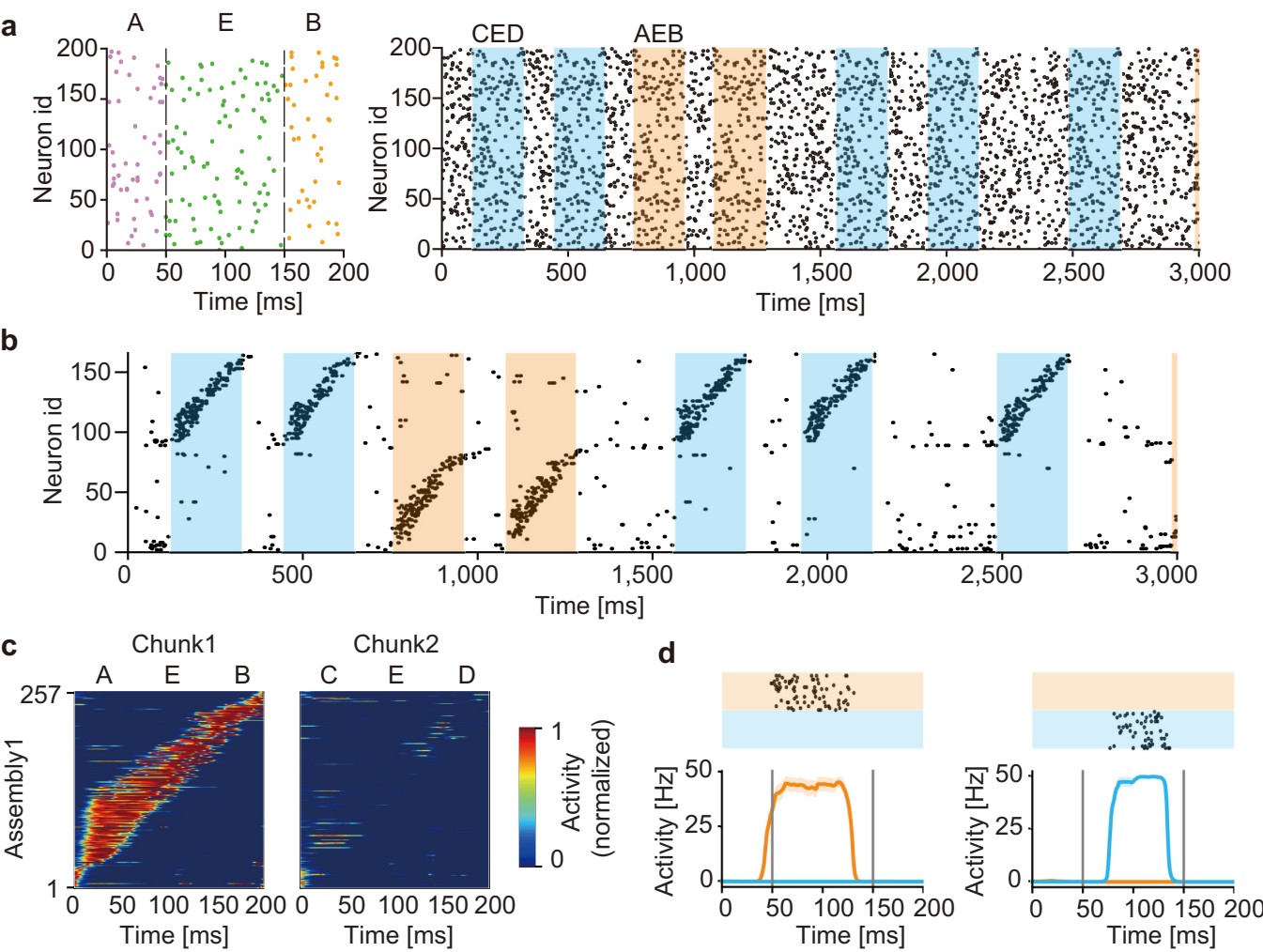

**Fig 2. Learning of overlapping chunks.** (a) Two chunks "AEB" (orange shade) and "CED" (blue shade) were repeated in input Poisson spike trains (left). The chunks were separated by random spike trains with variable lengths of 50 to 400 ms (unshaded). All neurons had the same firing rate of 5 Hz. Example spike trains during the initial 3 seconds are shown (right). In each chunk, the component patterns "A", "B", "C", and "D" were 50 ms-long and the shared component "E" was 100 ms-long. (b) Output spike trains of the trained network model. Neurons were sorted according to their onset response times, and only 160 out of the total 500 neurons are shown for the visualization purpose. A selective cell assembly emerged for each chunk. (c) The average responses of "AEB"-selective assembly to chunks "AEB" (left) and "CED" (right) are shown. The responses were averaged over 20 presentations of the chunks and normalized by the maximal response to the preferred chunk (i.e., chunk 1). (d) Responses of a "AEB"-selective neuron (left) and a "CED"-selective neuron in the trained network are shown. The raster plots show responses over 20 trials.

learning, we investigated how the somatic and dendritic compartments and gating factors of individual neurons behave during training. The dendritic compartments of these neurons responded to a preferred component pattern irrespective of which chunk the component appeared, showing that the dendrites were unable to discriminate the same component pattern as shared by different chunks (Fig 3C). In contrast, the gating factors responded differently to the same component pattern appearing in different chunks depending on the preceding component patterns (Fig 3D). This selective gating is thought to arise from a memory effect which is generated by recurrent synaptic input and determined by the previous state of the network. As a consequence, the somatic compartment could selectively respond to a particular chunk that strongly activated the gating factor during the presentation of the preferred component pattern (Fig 3E). Similar results are shown for other neurons in the trained network (S3C Fig).

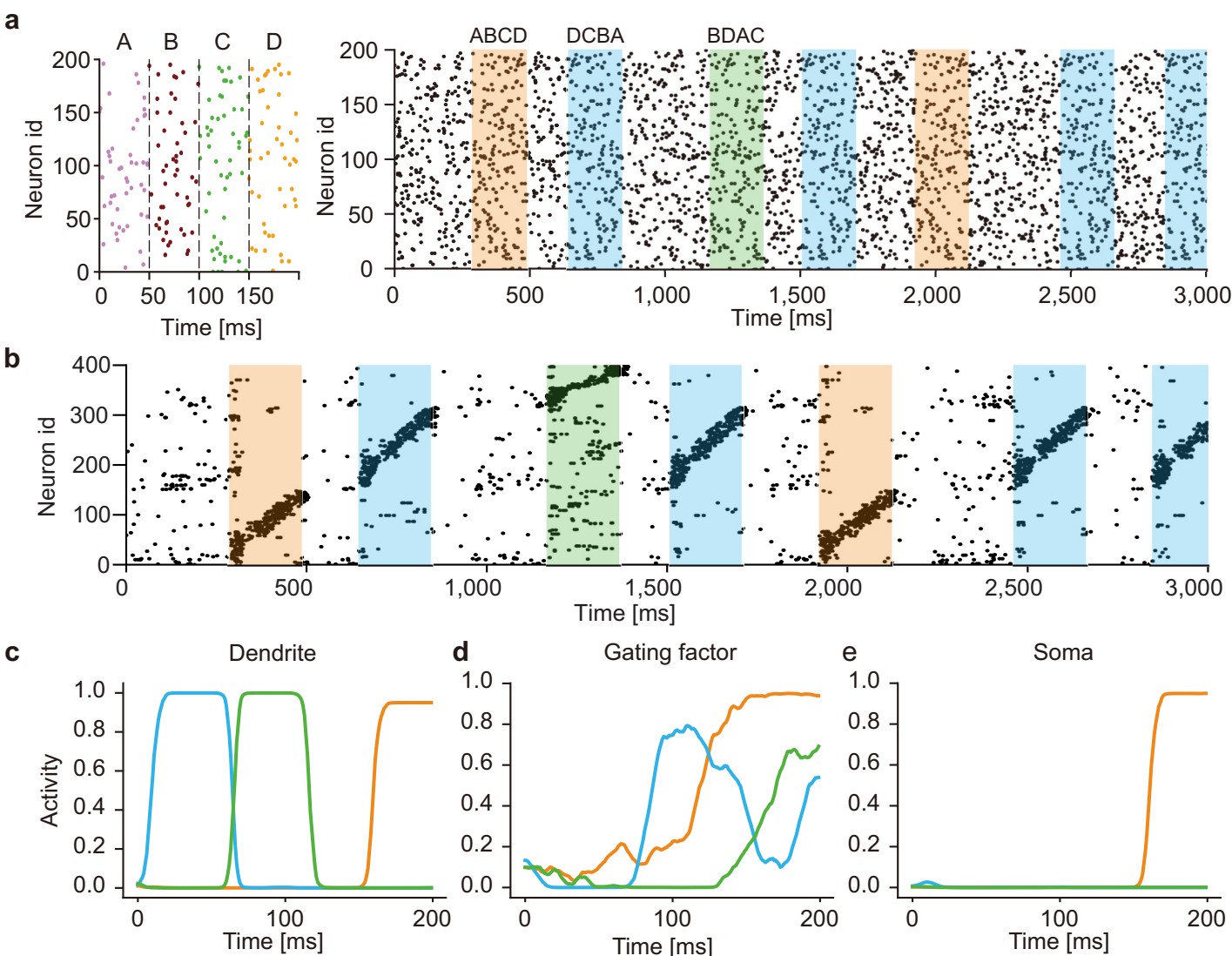

**Fig 3. Learning of complex spike sequences with recurrent gating.** (a) Three chunks were presented, which consists of four identical component patterns with its specific order (left). The example input spike train during first three seconds are shown (right). The regions filled in orange, blue and green represent the times when chunks "ABCD", "DCBA" and "BDAC" were presented, respectively. All neurons generate Poisson spike with the constant firing rate of 5Hz. Random spike sequences were presented in the unshaded areas. (b) Output spike trains of the trained network are shown. Neurons were sorted according to their onset response times, and only 400 out of 1200 neurons are shown for the visualization purpose. (c-e) The time evolution of dendritic activity, gating factor, and somatic response are shown in a neuron. Orange, blue and green traces represent the responses to its preferred component pattern "D" when the corresponding chunks were presented. The activities were averaged over 20 trials.

To explore the mechanism of the context dependent computation in our model, we analyzed the structure of recurrent connections in the trained model. Here, we first classified network neurons into 12 sub-groups according to the selectivity for the 4 component patterns of the 3 chunks. The average values of the weights between each group were then calculated to quantify the average strength of gating recurrent connections between the subgroups (S4A Fig). The diagonal elements of the matrix are greater than the non-diagonal elements, suggesting that neurons within the same sub-groups are connected by strong excitatory synapses. A further analysis of the interactions between the different assemblies (S4B Fig) shows each assembly has excitatory connections to the assembly corresponding to the next component in

the same chunk (e.g., "A" to "B" in chunk 1), while inhibitory connections to the assemblies for the previous component (e.g., "B" to "A" in chunk 1). Further, strong inhibition was formed among the assemblies selective to the same letters in different chunks (e.g., "B" in chunk 1 and "B" in chunk 2). Thus, a structured connectivity consistent with sequence structure emerges from the context-dependent learning.

The single-cell model proposed previously [19] could not perfectly segment similar sequence patterns as shown in Figs 2 and 3. Though the consistency between somatic and dendritic activities forces the neuron to respond to a specific input pattern, it could not perfectly discriminate similar patterns involving overlapping components. In the recurrent gating network, recurrent input on each neuron selectively passes one of similar input patterns from the dendrite to the soma, enabling the context-dependent segmentation. This cooperative function of recurrent synaptic input is difficult to prove analytically but is understandable because both afferent and recurrent synapses are trained by the same learning rule to achieve the same goal, that is, an optimally predictable somatic response.

Like other learning models, the performance of the proposed model varies depending on the values of multiple parameters. We evaluated the trainable number of sequences for a given size of the network. This number was larger in a network with 600 neurons than in that of 300 neurons, indicating that performance in learning is degraded as the number of input sequences is increased (S5A Fig). Second, we measured how the performance depends on the parameter $\gamma$, which determines the hysteresis effects in the mean and variance of past neuronal activities (Eq (5)-(6) in Methods). As we have shown previously [19], standardization with moments of membrane potentials is crucial for avoiding a trivial solution. We found that the performance of learning is deteriorated as the parameter $\gamma$ is increased (S5B Fig). This result is reasonable as a larger value of $\gamma$ weakens the hysteresis effect, making the standardization less stable. Finally, we explored to what extent the strength of static recurrent inhibitory connections $J/\sqrt{N}$ affect the model performance. We measured the performance at various values of the scaling parameter $J$ and found the best performance at $J = 0.5$ (S5C Fig). These results show that the model does not require a fine tuning of parameters as the performance is not narrowly peaked.

Our model developed low-dimensional representations that strongly reflect the temporal structures of chunks. The principal component analysis (PCA) of the network responses to the overlapping chunks shown in Fig 2 revealed that a smaller number of eigenvectors explained a larger cumulative variance as the training progressed (Fig 4A). At different stages of learning, the low-dimensional trajectories differently represented the chunks. Before learning, the two chunks and unstructured input segments (i.e., random spike trains) occupied almost the same portions of the low-dimensional trajectories (Fig 4B, left). At the mid stage of learning, the portions of the chunks grew while those of the random segments shrank (Fig 4B, middle). Neural states evolved along separate trajectories at the initial (corresponding to "A" and "C") and final (corresponding to "B" and "D") parts of the chunk-representing portions whereas the middle part (corresponding to "E") was tangled. After sufficient learning, the trajectories were completely separated (Fig 4B, right). As previously shown, recurrent gating crucially contributed to this separation. Indeed, the network generated almost overlapping trajectories for the pattern "E" if we fastened recurrent gating during learning and test (Fig 4C) or if we trained the model with recurrent gating but fastened it during test (Fig 4D). Two trajectories for the overlapping chunks were not clearly separable due to large fluctuations if we randomly shuffled the learned recurrent connections to destroy their connectivity pattern (Fig 4E). Thus, the context-dependent gating depends crucially on the learned fine structure of recurrent connections.

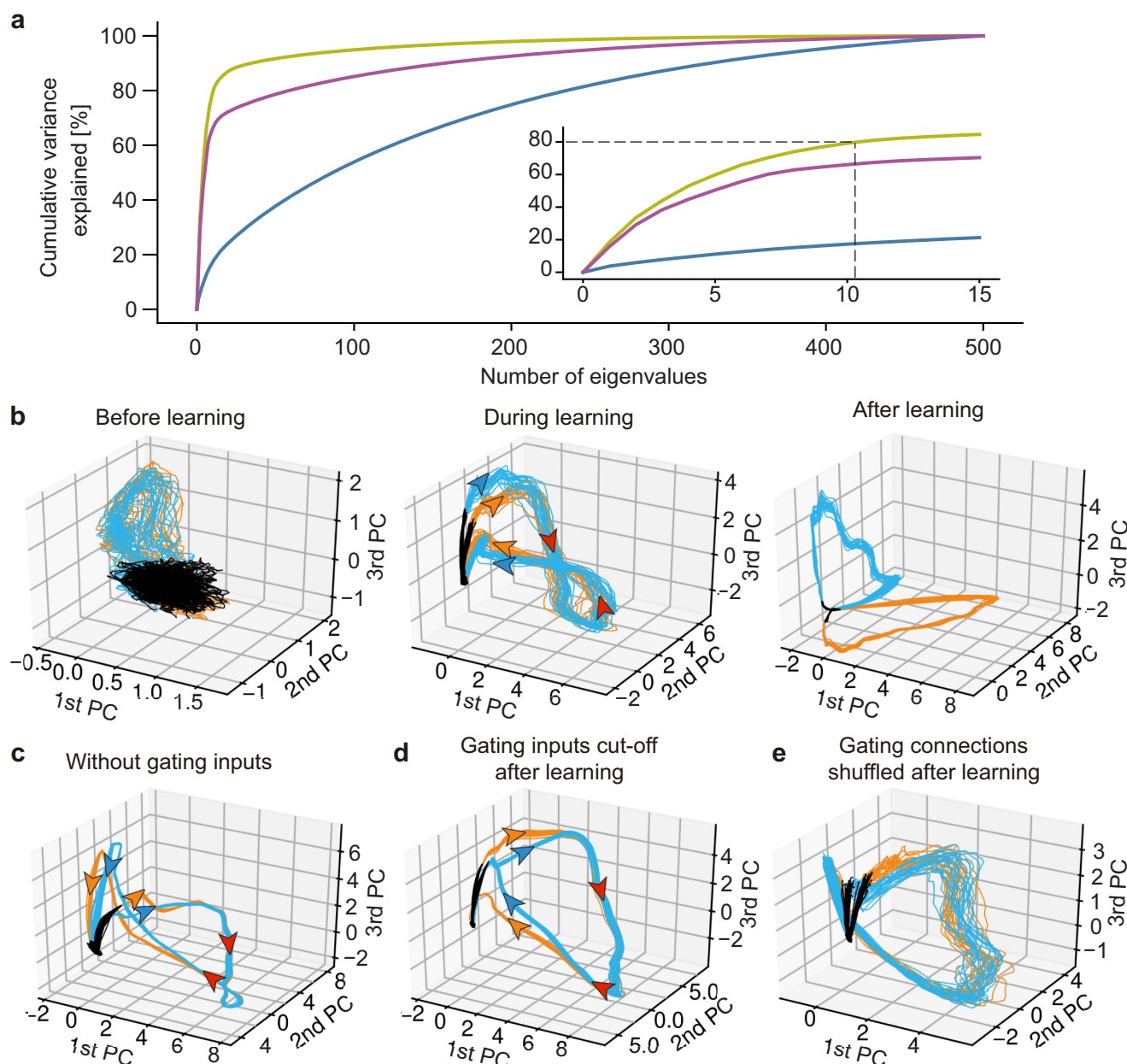

**Fig 4. Principal component analysis of the trained network.** (a) Cumulative variance explained of the PCs of the activities of before (blue), during (purple), and after (yellow) learning. Inset is an expanded view for major eigenstates. (b) The PCA-projected trajectories of network activity before, during, and after training are shown in the space spanned by PC1 to PC3. The network was trained with the same task as in Fig 2. The black, orange, and blue trajectories represent the periods during which random spike input, chunk 1 and chunk 2 were presented, respectively. (c) Recurrent gating was fixed in all neurons during the whole simulation. (d) The gating factor was clamped after the network in (b) were trained. (e) Recurrent connections were randomly shuffled after the network in (b) were trained. The directions of state evolution are indicated by arrows along the PCA-projected trajectories. The arrows are colored according to the corresponding chunks on the separated parts of the trajectories while they are colored in red on unseparated parts.

While the network model could learn noisy chunks as far as jitters in spike times were not too large (S6A and S6B Fig), the magnitude of jitters strongly influenced learning speed. This was indicated by the slow saturation of normalized mutual information (see Methods) between network responses and the true labels of chunks during learning (S6C Fig). The normalized mutual information took near the maximum value ($\approx 1$) as far as the variance of jitters fell

within the length of chunks (50 ms). This information dropped rapidly beyond the chunk length (S6D Fig).

Just like the brain can recognize a learned sequence irrespective of the length of its presentation, a learned pattern is detectable for the network even if the pattern is presented with a length different from the learned one (Fig 5A and 5B). We quantified the similarity of network responses to otherwise the same input patterns with different lengths. By calculating the rank-order correlations between responses to stimuli presented with three different lengths, we measured to what extent the serial order of neural responses are preserved over different conditions. In the network that learned the original pattern, the similarity increased significantly for all three durations of stimulus presentation (Fig 5C), suggesting that our model learns the manifold of temporal spike patterns rather than individual specific patterns. We examined the relative duration (RD) of input sequence beyond which the gating and non-gating models lose the ability of recognizing a learned sequence. The spike pattern used in Fig 3 was used. As shown in Fig 5D, performance was gradually deteriorated in both models as the RD was increased. While the gating model showed better performance than the non-gating model for RDs less than 3, the superiority of the gating model disappeared when the RD reached 4. These results demonstrate that the gating model recognizes sequences with RDs up to about 3. The robustness shown above raises a question about whether the present model can discriminate precise temporal spike patterns. Indeed, the network model clearly discriminated between similar but different input patterns when the inputs were learned as separate chunks. To study this, we trained the network model with random spike trains involving a repeated temporal pattern and stimulated the learned model with the original pattern (Fig 5E, top) and its time-reversal version (Fig 5E, bottom). The cell assembly that only learned the original pattern also responded to the reversed pattern in a reversed temporal order, meaning that the different temporal patterns were not discriminable in this case (Fig 5F). Interestingly, the same network model trained with both original pattern and time-reversed pattern self-organized distinct cell assemblies selective for the individual patterns (Fig 5G). This result may account for discrimination between "break" and "brake" when these words were learned as separate entities.

## Cell assembly detection in large-scale calcium imaging data

A virtue of our model is its applicability to analyzing large-scale neural recoding data. We show this in two calcium imaging data. The first data contains the activities of 452 hippocampal CA1 neurons recorded from mice running back and forth along a linear track between two rewarded sites (Fig 6A, top) [23]. Repetitive sequential activations of place cells were reported previously in the data (https://github.com/zivlab/island). For the use of our model, we binarized the data by thresholding activity of each neuron at the 50% of its maximal intensity (Fig 6A, bottom). After training, model neurons detected groups of input spike trains that tended to arrive in sequences, each of which was preferentially observed at a particular position of the track in a particular direction of run (Fig 6B). Sorting the activities of hippocampal neurons according to the sequential firing of model neurons (Methods) revealed place-cell sequences without referring to the behavioral data (Fig 6C).

Our second example is from the visual cortex in mice running on an air-floating ball [24]. The 525 second-long dataset [25] contains the activity of 6,532 neurons recorded by two-photon calcium imaging from the visual cortex as well as the behavioral data (running speed, pupil area, and whisking) monitored simultaneously with an infrared camera (Fig 6D) (https://figshare.com/articles/dataset/Recordings_of_ten_thousand_neurons_in_visual_cortex_during_spontaneous_behaviors/6163622/4). Due to the large data size, detecting cell assemblies is computationally challenging in this dataset. After training, the network model formed

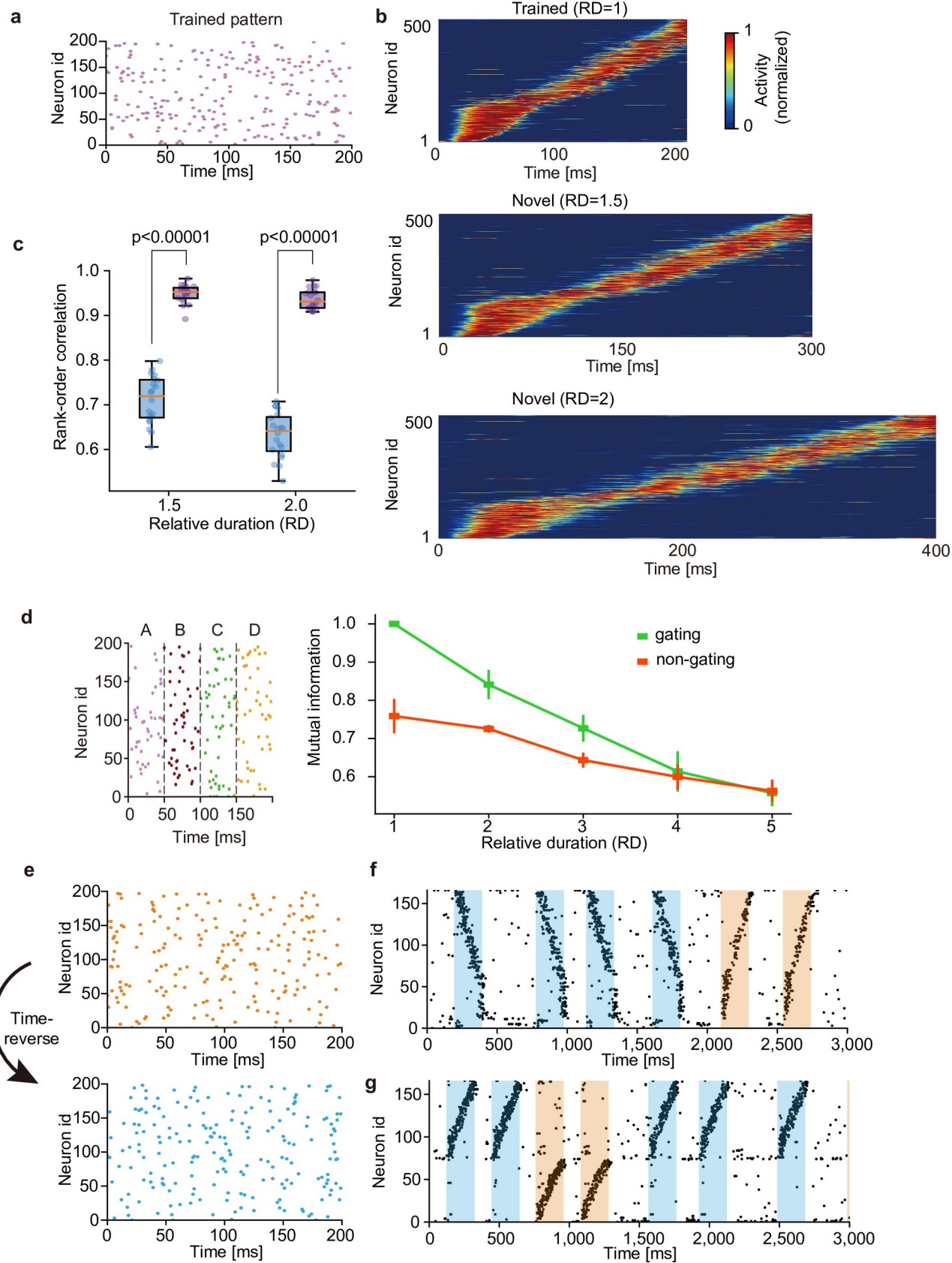

**Fig 5. Context-dependent learning of sequence information.** (a) For testing on time-warped patterns, the network was trained on random spike trains embedding a single pattern. (b) The trained network responded sequentially to the original and stretched patterns with two untrained lengths (i.e., the relative durations RD of 1.5 and 2). (c) Similarities of sequential order between the responses to the original and two untrained patterns were measured before (blue) and after (purple) learning. Independent simulations were performed 20 times, and p-values were calculated by two-sided Welch's t-test. (d) The input spike pattern used in the task in Fig 3 was considered (left) to quantify the degree of time warping that can be tolerated by our network model. Performance of both gating and non-gating models trained by input patterns with various relative durations are shown. (e) A time-inverted spike pattern (bottom) was generated from a original pattern (top). (f) The network was exposed to the original pattern in (e) during learning, and its responses were tested after learning for both original and time-inverted patterns. Both patterns activated a single cell assembly. (g) The network was exposed to both original and time-inverted patterns during learning as well as testing. Two assemblies with different preferred patterns were formed. For the visualization purpose, only 160 out of 500 neurons are shown in (f) and (g).

several neural ensembles, each of which displayed distinct spatiotemporal response patterns (Fig 6E). Interestingly, these neural ensembles showed their maximal responses at different periods of time, and the pupil area also changed its maximal size depending on active neural ensembles (Fig 6D and 6E). By sorting cortical neurons according to the response patterns of model neurons (Methods), we could find the repetition of distinct cell assemblies in the visual cortex (Fig 6F). The result revealed that active cell assemblies were changed between the early ($< 280$–$290$ s) and late epoch of spontaneous behavior, despite that there was no clear distinction in behavior (S7 Fig). To find the cell-assembly structures, we grouped co-activated model neurons (Fig 6G: see Methods). Unexpectedly, the time necessary for learning did not change much with data size, or the time was even slightly shorter for larger data sizes (Fig 6H). Presumably, this unintuitive result was because each cell assembly was represented by more neurons in larger data [19].

As the gating mechanism is crucial for context-dependent computation, we studied the effect of gating on the analysis of neural recording data. We first applied the model without recurrent gating (this model is equivalent to the previous feedforward network [19]) to the analysis of hippocampal CA1 data. Interestingly, without recurrent gating, the model could not separate the two sequences corresponding to forward and backward runs (S8A Fig). Next, we applied the non-gating model to the data recorded from the mouse visual cortex. The model showed structured activity patterns (S8B Fig) and detected cell assemblies (S8C Fig). However, compared to the gating model, the non-gating model generated monotonous responses, suggesting that the cell assemblies detected were contaminated (S8D Fig). Further, the responses of the gating model were significantly more correlated with the various behaviors of mice than those of the non-gating model (S8E Fig). These results show the crucial contribution of recurrent gating to cell assembly detection.

## Redundant information representations on dendrites

Evidence from the visual cortex [26], retrosplenial cortex [27], and hippocampus [28] suggested that the representations of sensory and environmental information in cortical neurons are more redundant on the dendrites compared to the soma. The dendrites can have multiple receptive fields while the soma generally represents only one of these receptive fields. The soma is likely to access information represented in a subset of the dendritic branches that share the same receptive field. A similar redundant coding occurs in the present somato-dendritic sequence learning.

To show this, we constructed a recurrent network of neurons having three dendritic components and simulated how the model learns orientation tuning. The individual dendritic branches were assumed to undergo independent recurrent gating and mutual competition through softmax (see Methods) (Fig 7A). We repeatedly presented a 40 ms-long random sequence of noisy binary images of oriented bars every 80 ms. The size of each image was 28×28 (= 784) pixels and each bar has a width of 7 pixels. Each pixel was flipped with the

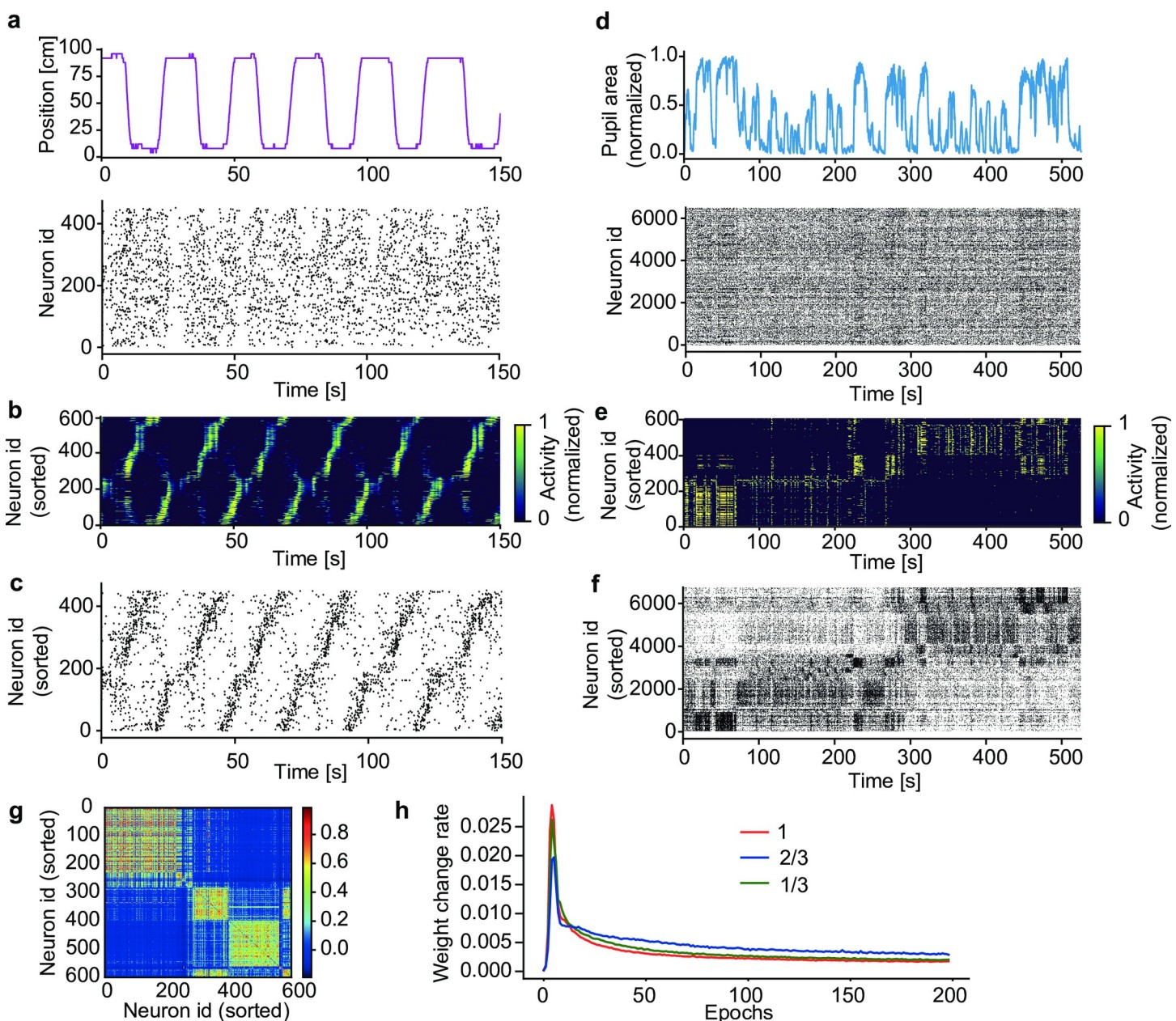

**Fig 6. Detecting salient activity patterns in calcium imaging data.** (a) The positions of a mouse (top) on a linear track and calcium imaging data of activity of 452 hippocampal CA1 neurons (bottom) were obtained from previously recorded data [23]. (b) The learned activities of model neurons were sorted according to their onset response times. (c) Each CA1 neuron was associated with a model neuron having the highest mutual correlation with the CA1 neuron. Then, the CA1 neurons were sorted according to the serial order of model neurons shown in (b). (d) The time course of normalized pupil area (top) and simultaneously recorded activities of 6,532 visual cortical neurons (bottom) were calculated from previously recorded data [24,25]. (e) Activity of a trained network model was sorted according to their onset response times (Methods). (f) Activities of the cortical neurons were sorted as in (c). (g) Correlation matrix of the population of network neurons is shown. (h) Learning curves over 200 epochs for various size of input neurons are shown. Red, blue and green traces show learning curves with the number of input neurons 1, 2/3, 1/3 times smaller than original 6,532 neurons. The weight change rate was calculated as the ratio of the sum of the absolute values of synaptic changes to the sum of the absolute values of all synapses.

probability of 0.1, and circular mass was applied to the images to suppress artifacts from the edges. Input neurons encoded the current value of a pixel by firing with 10 Hz.

During learning, the competition suppressed the dendritic activities that were less correlated with the somatic responses. In the self-organized network, the somatic compartments

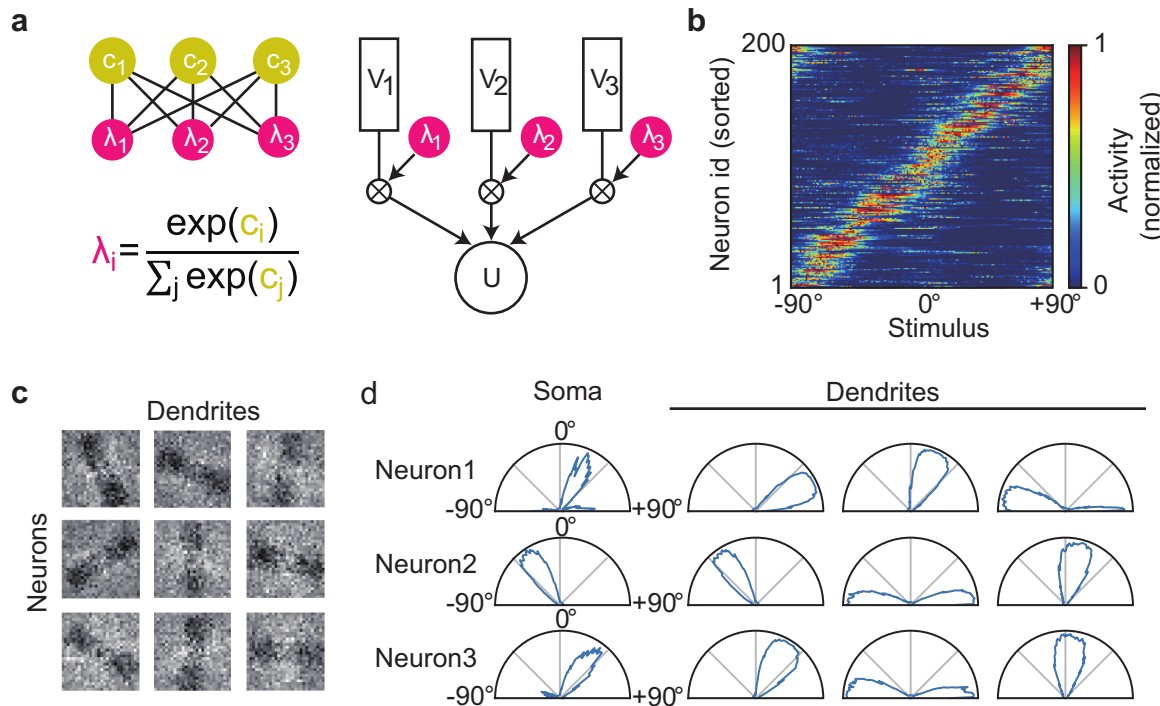

**Fig 7. Redundant dendritic representations of preferred sensory features.** (a) A schematic illustration of the neuron model with three dendritic compartments. The dendritic branches have independent gating factors, which compete with each other by softmax. (b) Somatic responses are shown for all neurons in the trained network. (c) Trained weight matrices are displayed for afferent inputs to three dendritic branches of three example neurons. (d) Somatic and dendritic activities of the three neurons in (c) are shown.

acquired unique preferred orientations (Fig 7B). In contrast, dendritic branches displayed different preferred orientations in some neurons (Fig 7C and 7D). Such redundant representations were not found in neurons having three dendritic components without gating (S9 Fig). Thus, the learning rule and recurrent gating proposed in this study possibly underlie the somatic selection process of redundant dendritic representations.

## Role of the conventional recurrent synaptic input

While the multiplicative recurrent input (i.e., recurrent gating) is crucial for segregating complex chunks, what is the role of additive (i.e., conventional) recurrent input? We demonstrate that the additive component is still needed for retrieving chunked sequences, namely, for pattern completion. We simulated a network model having both recurrent gating and non-vanishing additive recurrent inputs. Non-gating recurrent connections were trained with the same rule as for afferent connections. The network received a temporal input containing two mutually overlapping chunks (Fig 8A), and all synaptic connections underwent learning. The trained network formed two cell assemblies responding selectively to either of the chunks, as in the previous network without additive recurrent connections (S10 Fig). Since only additive recurrent input, but not recurrent gating, can activate postsynaptic cells, the additive input generates a reverberating activity, which may in turn assist the retrieval of learned sequences. This actually occurred in our simulations. Applying a cue stimulus, which was the first component pattern of one of the learned chunks, enabled the trained network to retrieve the subsequent component patterns in the chunk (Fig 8B and 8C). Thus, recurrent gating and additive recurrent inputs contribute to learning and retrieval of segmented sequence memory, respectively, in our network model.

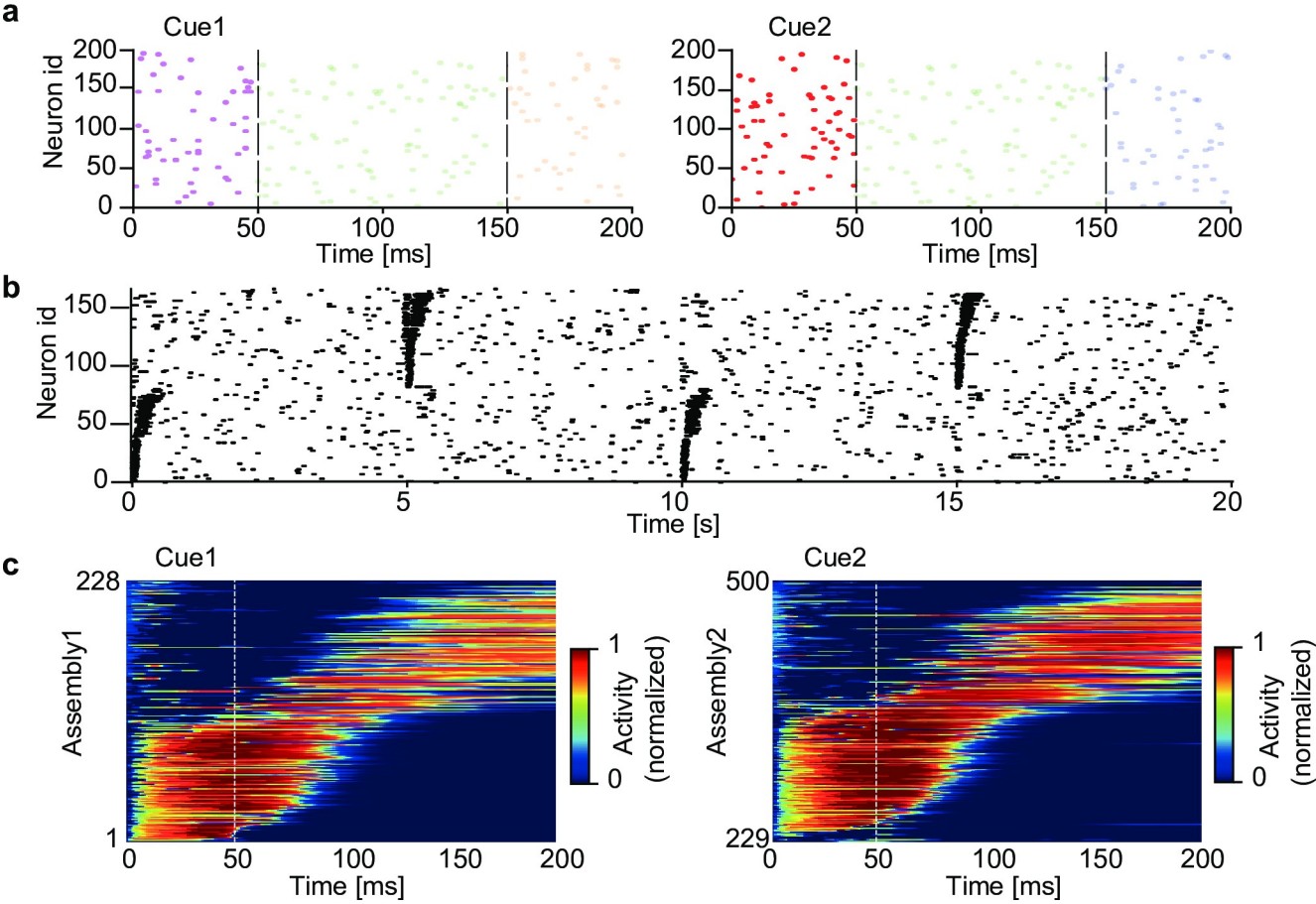

**Fig 8. Spontaneous completion of learned sequences.** The two chunks shown in Fig 2 were used for training a network having both recurrent gating and additive recurrent inputs. (a) In the testing phase, the first component pattern of each chunk (cue 1 or cue2; dark raster plots), but not the subsequent component patterns (light raster plots), was presented to the network. (b) The raster plot of network activities in the testing phase is shown. Cue 1 and cue 2 were presented alternately every 5 seconds. Neurons were sorted according to their onset response times, and only 160 out of the total 500 neurons are shown for the visualization purpose. (c) Sequential responses of the two assemblies were averaged over 20 trials. Vertical lines indicate the end of cue presentation. The sequential responses were evoked in the learned order.

## Discussion

In this study, we constructed a recurrent network of compartmentalized neuron models to explore the neural mechanisms to segment temporal input. The crucial role of recurrent gating in context-dependent chunking of complex sequences is a major finding. Recurrent gating enables the instantaneous network state to regulate the degree of the dendro-somatic information transfer in single neurons in different contexts. In contrast, simple segmentation tasks do not necessarily require recurrent connections. With the help of recurrent gating, the model is capable of detecting the fine structures of cell assemblies in large-scale neural recording data.

Our model describes a possible form of integrating dendritic computation into computation at the network level. Learning in our model minimizes the prediction error between the soma and dendrites, thus improving the consistency in responding to synaptic input between the input terminal (dendrites) and the output terminal (soma) of single neurons. This enables the neurons to learn repeated patterns in synaptic input in a self-supervised manner. Previous theoretical studies utilized the local dendritic potential with a fixed gating factor to predict the somatic spike responses [19,22]. We extended the previous learning rule over recurrent

connections such that recurrent input helps the dendritic compartment to predict the somatic responses by regulating the degree of signal transfer to the soma in a network state-dependent manner. As a consequence of recurrent gating, the soma can respond differently to the same sequence component depending on the preceding element in sequences, whereas the dendrites respond similarly to the same component. As in our model, some neurons in the premotor nucleus HVC in canaries change their responses to a song element depending on the preceding phrases in songs [13]. However, the response of HVC neurons can also vary according to the following phrases in songs. Such response modulations are likely to represent action planning, which was not considered in this study. Previous experimental and theoretical studies suggested that dendritic inhibition implements a gating operation on synaptic input [29–32]. The role of inhibition on the context-dependent segmentation of input should also be investigated further.

Previously, spike-timing-dependent plasticity was used for detecting recurring patterns in input spike trains in a recurrent neural network without recurrent gating [18]. While the model successfully discriminated relatively simple sequences, it could not discriminate complex sequences involving, for instance, overlapping spatiotemporal patterns. Our results suggest that additive recurrent connections are unlikely to be crucial for learning hierarchically organized sequences. These connections are necessary for retrieving chunked sequences but are unnecessary for learning these sequences. Our results instead suggest that such learning crucially relies on recurrent gating and its context-dependent tunning. Thus, multiplicative and additive recurrent connections have a clear division of labor in the present model. Recurrent synapses were shown to amplify the responses of cortical neurons having similar receptive fields [33], and this amplification resembles the selective amplification of a particular sequence component shown in this study. Further, it was previously shown that postsynaptic inhibition multiplicatively modulates the membrane potential of visual neurons in the locust [34]. In addition, a recent study suggests that the thalamocortical feedback is necessary for reliable propagation of a synaptically evoked dendritic depolarization to the soma in layer-5 pyramidal neurons [35]. This phenomenon is reminiscent of the gating mechanism proposed in this study. Thus, such a mechanism may be realized on a large spatial scale by cortico-thalamocortical recurrent circuits. However, the biological mechanisms of recurrent gating are open to future studies.

Some neural network models in artificial intelligence also utilize gating operations. A well-known example is Long Short-Term Memory (LSTM) for sequence learning and control [36]. The most general form of LSTM contains three types of gate functions, i.e., input, output, and forget gates, and these functions are optimized through supervised learning. In an interesting attempt at the learning-to-learn paradigm [37–39], LSTM was coupled with another network model for self-supervised learning of visual features [40]. In another LSTM-inspired model, neural dynamics with oscillatory gated recurrent input were used to convert spatial activity patterns to temporal sequences in working memory and motor control [21]. In contrast to LSTM, our model learns the optimal gating of the dendro-somatic information transfer by seeking a self-consistent solution to the optimization problem without supervision. Unlike in LSTM, our model only has a single type of gate, which likely corresponds to the input gate of LSTM, to regulate a current flow into the output terminal (soma) of the neuron. In LSTM, however, the most influential gate on learning performance is thought to be the forget gate [41,42]. It is intriguing to ask whether and how recurrent networks learn an optimal forget gate for unsupervised learning of hierarchical sequences in addition to the proposed gate.

When multiple dendritic branches compete for repeated patterns of synaptic input to a single neuron, recurrent gating enables these branches to learn different input features. Consequently, in each neuron, the dendrites learn more redundant representations of input

information than the soma. In pyramidal neurons in the rodent primary visual cortex, the dendritic branches have heterogeneous orientation preferences while the somata have unique orientation preferences [26]. Similarly, in retrosplenial cortex [27] and place cells in the hippocampal CA3 [28], the dendritic branches have multiple receptive fields whereas the somata have unique receptive fields. Our model provides a possible neural mechanism for these redundant representations on the dendrites. When the environment suddenly changes, such redundancy may allow neural networks to quickly remodel their responses to adapt to the novel situation. However, the functional benefit of this redundancy has yet to be clarified.

A practically interesting feature of our model is its applicability to large-scale neural recording data. For such purposes, various mathematical tools have been proposed based on methods in computer science and machine learning [43–47]. However, many of these methods suffer time-consuming, combinatorial problems necessary for an exhaustive search for activity patterns in the neural population. In contrast, our model with a biologically inspired learning rule is free from this problem, presumably due to the same reason that cortical circuits do not have this problem. Actually, the present data from the mice visual cortex contain more than 6,000 active neurons, yet our analysis revealed clear evidence for cell assembly structures. These results are interesting because they suggest that cell assemblies underlie the multidimensional neural representations of mice spontaneous behavior [24]. As the size of neural recording data is increasing rapidly, the low computational burden and high sensitivity to structured activity patterns show big advantages of this model.

## Methods

### Neural network model

Our network model consists of $N_{in}$ input neurons and $N$ recurrently connected neurons. Each neuron in the recurrent network consists of two compartments: the somatic and dendritic compartments. Inspired from a previous single neuron model, the somatic response can be approximated as an attenuated version of the dendritic potential $V$ [22]. In our recurrent network model, the dendro-somatic signal transfer is regulated by the gating factor $\lambda$ that depends on recurrent synaptic inputs through the local potential $c$ as follows:

$$c_i(t) = \mathbf{w}_i^{net(c)} \cdot \boldsymbol{e}^{net}(t), \tag{1}$$

$$\lambda_i(t) = g_G(\hat{c}_i(t)) \tag{2}$$

$$V_i(t) = \mathbf{w}_i^{net(V)} \cdot \boldsymbol{e}^{net}(t) + \mathbf{w}_i^{ext(V)} \cdot \boldsymbol{e}^{ext}(t), \tag{3}$$

where the subscript $i$ is the neuron index, $\mathbf{w}_i^{net(c)}$ are the $N$-dimensional weight vector of recurrent gating on the local potential $c$, and $\mathbf{w}_i^{net(V)}$ the $N$-dimensional weight vector of additive recurrent connections on the dendrite of the $i$-th neuron. In Eq (2), $g_G$ and $\hat{c}$ will be defined later. The $N_{in}$-dimensional vector $\mathbf{w}_i^{ext(V)}$ represents the weights of afferent inputs. Except in Fig 8, we set as $\mathbf{w}_i^{net(V)} = \mathbf{0}$. The variables $\boldsymbol{e}^{net}$ and $\boldsymbol{e}^{ext}$ are the post-synaptic potentials evoked by recurrent and afferent inputs, respectively. The initial values of $\mathbf{w}_i^{net(c,V)}$ and $\mathbf{w}_i^{ext(V)}$ were generated by Gaussian distributions with zero mean and the standard deviations of $1/\sqrt{N}$ and $1/\sqrt{N_{in}}$, respectively. All three types of connections are fully connected.

The dynamics of the somatic membrane potential are described as

$$\dot{U}_i(t) = -\frac{1}{\tau} U_i(t) + \lambda_i(t)(-U_i(t) + \hat{V}_i(t)) - \boldsymbol{G}_i \cdot \boldsymbol{e}^{net}(t), \tag{4}$$

where $\tau = 15$ ms is the membrane time constant. The last term in Eq (4) represents a peri-somatic recurrent inhibition with uniform inhibitory weights of the strength $J/\sqrt{N}$, with $J = 0.5$ in all simulations. No self-inhibition is considered. Further, $\hat{c}_i$ and $\hat{V}_i$ are the standardized potentials calculated as

$$\hat{c}_i(t) = [c_i(t) - \mu_i^c(t)]/\sqrt{\rho_i^c(t) - \mu_i^c(t)^2},\qquad(5)$$

$$\hat{V}_i(t) = [V_i(t) - \mu_i^V(t)]/\sqrt{\rho_i^V(t) - \mu_i^V(t)^2},\qquad(6)$$

where $\mu_i^c(t)$ and $\rho_i^c(t)$ are exponentially decaying averages of the membrane potential and its square of the gating compartment,

$$\mu_i^c(t) = (1 - \gamma)\mu_i^c(t - 1) + \gamma c_i(t),\qquad(7)$$

$$\rho_i^c(t) = (1 - \gamma)\rho_i^c(t - 1) + \gamma c_i(t)^2,\qquad(8)$$

respectively ($0 < \gamma < 1$). The values of $\mu_i^V(t)$ and $\rho_i^V(t)$ are calculated from $V_i$ in a similar fashion. As we have shown previously [19], the standardization enables the model to avoid a trivial solution. Without standardization, our learning rule can minimize the error between the somatic and dendritic activities to zero by making both activities simultaneously zero (see Eq (18)). This trivial solution occurs when all synaptic weights vanish after learning. The standardization prevents the trivial solution by maintaining temporal fluctuations of O(1) in the somatic membrane potential, thus ensuring successful learning of nontrivial temporal features. In Eq (2), the gating function $g_G(x)$ is defined as

$$g_G(x) = g_0[1 + \exp(-\beta_G(x - \theta_G))]^{-1},\qquad(9)$$

where $g_0 = 0.7$, $\beta_G = 5$ and $\theta_G = 0.5$.

The somatic compartment generates a Poisson spike train with saturating instantaneous firing rate given as

$$\phi(x) = \phi_0[1 + \exp(-\beta(x - \theta))]^{-1},\qquad(10)$$

where $\phi_0 = 0.05$ kHz, $\beta = 5$ and $\theta = 1$ throughout the present simulations.

Afferent inputs are described as Poisson spike trains of $N_{\text{in}}$ input neurons:

$$X_k^{\text{ext}}(t) = \sum_q \delta(t - t_{k,q}^{\text{ext}}),\qquad(11)$$

where $\delta$ is the Dirac's delta function and $t_{k,q}^{\text{ext}}$ is the time of the $q$-th spike generated by the $k$-th input neuron. The postsynaptic potential evoked by the $k$-th input is calculated as

$$\tau_s \dot{I}_k^{\text{ext}} = -I_k^{\text{ext}} + \frac{1}{\tau} X_k^{\text{ext}},\qquad(12)$$

$$\dot{e}_k^{\text{ext}} = -\frac{e_k^{\text{ext}}}{\tau} + e_0 I_k^{\text{ext}},\qquad(13)$$

where $\tau_s = 5$ ms and $e_0 = 25$. Note that the parameter $\tau$ in Eqs (12) and (13) is the membrane time constant used in Eq (4). The scaling by $\tau^{-1}$ ensures that the last term in Eq (12) is in a unit of current. The postsynaptic potentials induced by recurrent inputs, $e^{\text{net}}$, are similarly calculated.

### The optimal learning rule for recurrent gated neural networks

We derive an optimal learning rule for the gating recurrent neural network in the spirit of minimization of regularized information loss (MRIL), which we recently proposed for single neurons [19]. The objective function is the KL-divergence between two Poisson distributions associated with the somatic and dendritic activities:

$$E(\mathrm{W^V W^C}) = \left\langle \int_0^T dt \sum_i D_{KL}[(U_i(t))||\phi(V_i^*(t))] \right\rangle, \tag{14}$$

where angle bracket stands for the averaging over input spike trains, and $\mathbf{W}^\mathrm{V}$ and $\mathbf{W}^\mathrm{c}$ are the weight matrix of synaptic inputs onto the dendrite $V$ and those onto the local potential $c$ for recurrent gating, respectively. The gated dendritic potential $V_i^*$ is defined as

$$V_i^*(t) \equiv \frac{g_\mathrm{G}(c_i)}{g_\mathrm{L} + g_\mathrm{G}(c_i)} V_i(t), \tag{15}$$

where $g_\mathrm{L} = \tau^{-1}$. The crucial point in Eq (15) is that the degree of gating depends on $c$, and hence on network states through Eq (1).

The weights of all synaptic connections on the dendritic compartment (i.e., the weights of both afferent input and additive recurrent input) obey learning rules similar to the previously derived rule [19] except that the degree of gating is no longer constant in the present model:

$$\Delta\mathbf{w}_i^{\mathrm{net(V),ext(V)}} \propto -\frac{\partial E}{\partial \mathbf{w}_i^{\mathrm{net(V),ext(V)}}}$$
$$= \langle \textstyle\int_0^T dt\, \psi^\mathrm{V}(c_i, V_i^*)[\phi(U) - \phi(V_i^*)]\boldsymbol{e}^{\mathrm{net,ext}} \rangle, \tag{16}$$

where the function $\psi^\mathrm{V}(c_i, V_i^*)$ is defined as

$$\psi^\mathrm{V}(c_i, V_i^*) = \frac{\beta g_\mathrm{G}(c_i)}{g_\mathrm{L} + g_\mathrm{G}(c_i)}\left(1 - \frac{\phi(V_i^*)}{\phi_0}\right). \tag{17}$$

The learning rule for recurrent gating is novel and can be calculated by a gradient descent as follows:

$$\Delta\mathbf{w}_i^{\mathrm{net(c)}} \propto -\frac{\partial E}{\partial \mathbf{w}_i^{\mathrm{net(c)}}}$$
$$= \left\langle \int_0^T dt \left[ \phi(U_i)\frac{\partial}{\partial \mathbf{w}_i^{\mathrm{net(c)}}}\log\phi(V_i^*) - \frac{\partial}{\partial \mathbf{w}_i^{\mathrm{net(c)}}}\phi(V_i^*) \right] \right\rangle$$
$$= \left\langle \int_0^T dt \frac{\phi'(V_i^*)}{\phi(V_i^*)}[\phi(U_i) - \phi(V_i^*)] V_i \frac{\partial}{\partial \mathbf{w}_i^{\mathrm{net(c)}}}\frac{g_\mathrm{G}(c_i)}{g_\mathrm{L} + g_\mathrm{G}(c_i)} \right\rangle \tag{18}$$
$$= \left\langle \int_0^T dt \frac{\beta_\mathrm{G} g_\mathrm{L} g_\mathrm{G}(c_i)\left[1 - \frac{g_\mathrm{G}(c_i)}{g_0}\right]}{[g_\mathrm{L} + g_\mathrm{G}(c_i)]^2}\frac{\phi'(V_i^*)}{\phi(V_i^*)}[\phi(U_i) - \phi(V_i^*)]V_i\boldsymbol{e}^{\mathrm{net}} \right\rangle$$
$$= \langle \textstyle\int_0^T dt\, \psi^\mathrm{c}(c_i, V_i^*)[\phi(U_i) - \phi(V_i^*)]V_i\boldsymbol{e}^{\mathrm{net}} \rangle,$$

where the function $\psi^c(c_i, V_i^*)$ is defined as

$$\psi^c(c_i) = \frac{\beta_G g_L \left[1 - \frac{g_G(c_i)}{g_0}\right]}{g_L + g_G(c_i)} \psi^V(c_i). \tag{19}$$

In the present simulations, we used an online version of the above learning rules:

$$\Delta \mathbf{w}_i^{\text{net(V),ext(V)}} = \varepsilon^{\text{net(V),ext(V)}} \psi^V(c_i, V_i^*)[\phi(U_i) - \phi(V_i^*)]\boldsymbol{e}^{\text{net,ext}} \tag{20}$$

$$\Delta \mathbf{w}_i^{\text{net(c)}} = \varepsilon^{\text{net(c)}} \psi^c(c_i, V_i^*)[\phi(U_i) - \phi(V_i^*)]V_i \boldsymbol{e}^{\text{net}}, \tag{21}$$

where the learning rates were given as $\varepsilon^{\text{ext(V)}} = 10^{-5}$, $\varepsilon^{\text{net(V)}} = 10^{-5}$, and $\varepsilon^{\text{net(c)}} = 10^{-4}$.

## The optimal learning rule for multi-dendrite neuron model

For the multi-dendrite neuron model used in Fig 7, the membrane potential of the k-th dendrite of the i-th neuron and the dynamics of the corresponding somatic potential were calculated as

$$V_{i,k}(t) = \mathbf{w}_i^{\text{ext(V)}} \cdot \boldsymbol{e}^{\text{ext}}(t), \tag{22}$$

$$\dot{U}_i(t) = -\frac{1}{\tau}U_i(t) + \sum_{k=1}^K \lambda_{i,k}(t)(-U_i(t) + \hat{V}_{i,k}(t)) - \boldsymbol{G}_i \boldsymbol{e}^{\text{net}}(t), \tag{23}$$

where $K$ is the number of dendrites in each neuron. In this study, $K = 3$ for all neurons. We assumed that the dendritic compartments compete for the somatic activity of each neuron, governed by recurrent gating with a softmax function:

$$\lambda_{i,k}(\text{t}) = \frac{\exp(\beta_G c_{i,k}(t))}{\sum_l \exp(\beta_G c_{i,l}(t))}, \tag{24}$$

where $c_{i,k}$ is calculated as

$$c_{i,k}(t) = \mathbf{w}_{i,k}^{\text{net(c)}} \cdot \boldsymbol{e}^{\text{net}}(t). \tag{25}$$

Since $V_i^* = (g_L + 1)^{-1}\sum_l \lambda_{i,l} V_{i,l}$, it straightforward to derive the update rule for connections onto to the dendrites:

$$\Delta \mathbf{w}_{i,k}^{\text{ext(V)}} = \varepsilon^{\text{ext(V)}} \psi^V(\lambda_{i,k}, V_i^*)[\phi(U_i) - \phi(V_i^*)]\boldsymbol{e}^{\text{ext}}, \tag{26}$$

where

$$\psi^V(\lambda_{i,k}, V_i^*) = \frac{\lambda_{i,k}}{g_L + 1} \phi(V_i^*)\left(1 - \frac{\phi(V_i^*)}{\phi_0}\right). \tag{27}$$

Using the fact that $\partial \lambda_{i,k}/\partial c_{i,l} = \lambda_{i,k}(\delta_{l,k} - \lambda_{i,k})$, we can derive the update rule for recurrent gating as follows:

$$\Delta \mathbf{w}_{i,k}^{\text{net(c)}} = \varepsilon^{\text{net(c)}} \sum_l \psi_{k,l}^c(\lambda_i, V_i^*)[\phi(U_i) - \phi(V_i^*)]V_{i,l}\boldsymbol{e}^{\text{net}}, \tag{28}$$

where

$$\psi^c_{k,l}(\lambda_i, V^*_i) = (\delta_{l,k} - \lambda_{i,k})\psi^V(\lambda_{i,l}, V^*_i).$$ (29)

## Normalized mutual information score

In S3 Fig, we determined the estimated labels of the output response by Affinity Propagation [48], and then calculated the normalized mutual information score [49] between the estimated labels $X$ and the true label $Y$ as

$$\text{NMI} = 2\frac{I(X;Y)}{H(X) + H(Y)},$$ (30)

where $I(X;Y)$ is the mutual information between $X$ and $Y$ and $H(X)$ is the entropy of $X$.

## The Spearman's rank-order correlation

In Fig 5C, we quantified the extent to which the order of sequential responses was preserved in network activity. To this end, we calculated the Spearman's rank-order correlation [50] between network responses as

$$\rho = 1 - \frac{6\sum_{n=1}^{N} D_n^2}{N^3 - N},$$ (31)

where $N$ is the number of neurons in the network and $D_n$ is the difference in the ranks of the n-th neuron between two datasets when sorted according to their onset response times.

## The capacity of gating recurrent network

In S5A Fig, we considered the network of sizes 300 and 600. For each network, learning was performed with 3, 5, and 7 sequences. These sequences were randomly selected from 4! = 24 permutations of sequence a-b-c-d. After training, we evaluated the performance of the model by calculating the index $SI$ of feature selectivity as

$$SI = 1 - \frac{1}{K}\sum_{k=1}^{K}\frac{<r_k>_{\backslash k}}{<r_k>_k}$$

where $K$ is the number of stimuli (or the number of the corresponding assemblies) and $r_k$ is the population-averaged activity of the $k$-th assembly. Note that $0 \le SI \le 1$. The brackets $<r>_k$ and $<r>_{\backslash k}$ refer to temporal averages during the presentation of the $k$-th stimulus or all stimuli except the $k$-th stimulus, respectively.

## Neural sorting algorithms

In all figures except Fig 6E, neurons in the trained network were sorted according to the onset response times of these neurons. In Fig 6E, we first grouped neurons such that all pairs in a group had a correlation coefficient greater than 0.2. We then sorted the resultant groups based on their onset response times. In Fig 6C and 6F, we first sorted model neurons based on their peak response times. We then sorted the experimental data by associating each cortical neuron with a model neuron showing the highest correlation.

## Simulation parameters

The values of parameters used in the present simulations are as follows: in Figs 2, 4, 5, 8 and S1, S2, S6, and S10 Figs, $N = 500$, $N_{in} = 2,000$ and $\gamma = 0.0003$; in Figs 3 and 2. S3 and S4 Figs, $N = 1,200$, $N_{in} = 2,000$ and $\gamma = 0.0003$; in Fig 6A–6C and S4A Fig, $N = 600$, $N_{in} = 452$ and $\gamma = 0.0003$; in Fig 6D–6F and S8B and S8C Fig, $N = 600$, $N_{in} = 6,532$ and $\gamma = 0.00005$; in Fig 7 and S9 Fig, $N = 200$, $N_{in} = 28 \times 28$ and $\gamma = 0.0003$. Usually, the network was trained for the duration of 1,000 seconds. In Fig 6, the input spike trains constructed from experimental data were repeated 200 times during training.

## Data and Code

All numerical datasets necessary to replicate the results shown in this article can easily be generated by numerical simulations with the software code provided below. No datasets were generated during this study. All codes were written in Python3 with numpy 1.17.3 and scipy 0.18.1. Example program codes used for the present numerical simulations and data analysis are available at https://github.com/ToshitakeAsabuki/dendritic_gating.

## Supporting information

**S1 Fig. Learning curve of recurrent gating network.** Correlation coefficient between somatic and dendritic activity during learning in a task considered in Fig 2 is shown. Here, the 1,000 seconds-long learning period was divided into multiple training sections and the correlation coefficient between somatic and dendrite activity was calculated in each section. The solid line and shaded area (invisible) represent the mean and the s.d. of correlation over 10 independent simulations.
(PDF)

**S2 Fig. Learning of overlapping patterns in a recurrent network without gating.** (a) Output spike trains of the trained recurrent network are shown. Neurons were sorted according to their onset response times. (b) The responses to the two chunks were averaged over 20 trials. (c) PCA was applied to obtain the low-dimensional trajectories of the trained network. The black, orange, and blue portions indicate the periods of random spike input, chunk 1 and chunk 2, respectively. The two trajectories corresponding to the two chunks were inseparable and the network failed to learn the chunks.
(PDF)

**S3 Fig. Chunk-selective responses in the network trained in Fig 3.** (a) The responses of the first cell assembly to its preferred (left) and non-preferred (middle, right) chunks are shown. These responses were averaged and normalized as in Fig 2C. (b) Preferred responses of two neurons are shown as examples. Top and bottom traces show the responses of chunk2-D and chunk3-D selective neurons, respectively. (c) As in Fig 3C–3E, trial-averaged responses of dendrite, gating factor and soma are shown for two other neurons.
(PDF)

**S4 Fig. Analysis of trained gating recurrent connections.** (a) The average values of the weights between 12 assemblies defined according to the selective response to the 4 component patterns of the 3 chunks are shown. (b) Mean weights between groups of neurons in three cases are shown. Cyan and green colors indicate the connections project to assemblies correspond to previous and next component patterns within chunk, while magenta indicates inter-actions between assemblies correspond to the same component patterns but belong to

different chunks.
(PDF)

**S5 Fig. Model performance dependences on various parameters.** (a) Performances of networks with sizes 600 and 300 are shown over different number of chunks are shown. Error bars show s.d.s. (b) Learning performances of network of size 600 over various values of parameter γ (see Eq ([7]) and ([8])) are shown. Error bars show s.d.s. (c) Same as in (b), but over the strength of inhibition are shown.
(PDF)

**S6 Fig. Robustness against spike timing jitters.** (a) Responses of the networks trained on input spike trains with timing jitters of 70 ms (top) and 100 ms (bottom) are shown. Here, spike times within chunks were sifted by the amounts drawn by a Gaussian distribution with mean zero and s.d of jitter strength, and these jitters were present during learning and testing. Neurons were sorted according to the times of their response onsets during chunks, and only 160 out of the total 500 neurons are shown for the visualization purpose. (b) The normalized average activities of the two assemblies with timing jitters of 70 ms (left) and 100 ms (right) are shown. (c) Learning curves are shown when the average jitter was 0 ms (purple), 70 ms (green), and 100 ms (blue), respectively. The solid lines and shaded areas represent the averages and s.d over 20 trials, respectively. Learning performance was measured by the normalized mutual information between network activity and target labels (Methods). (d) The performance measures averaged over 20 trials are shown at various sizes of jitters. Error bars stand for the s.d.
(PDF)

**S7 Fig. Behaviors of freely behaving mouse.** Running speed (top), whisking (middle), and pupil area (bottom) of freely behaving mouse are shown.
(PDF)

**S8 Fig. Analysis of calcium imaging data without recurrent gating.** (a) The positions of a mouse on a linear track [23] (top) and the activities of model neurons learned without recurrent gating (bottom) are shown. Model neurons were sorted according to their onset response times. Separations between the two sequences corresponding to forward and backward runs are invisible (c.f. Fig 6B). (b) Activities of model neurons trained on the neural data recorded from the mice visual cortex [24,25] without recurrent gating are shown. The model neurons were sorted according to their onset response times. (c) We associated each cortical neuron with a model neuron having the highest correlation with the cortical neuron. Then, we sorted the cortical neurons according to the serial order of model neurons shown in (b). (d) Population-averaged activities of network model trained on the data of visual cortex with (left) and without (right) gating are shown. (e) Correlation between average activities shown in (d) and various behaviors are shown. Blue and magenta plots correspond to gating and non-gating, respectively. In both type of networks, 10 independent simulations were performed.
(PDF)

**S9 Fig. Multi-dendrite neuron model without gating.** (a) A schematic illustration of the neuron model with three dendritic compartments without gating. (b) Trained weight matrices are displayed for afferent inputs to three dendritic branches of three example neurons. (c) (d) Somatic and dendritic activities of the three neurons in (b) are shown.
(PDF)

**S10 Fig. Sequence learning in the copresence of recurrent gating and recurrent input.** Dendrites received additive recurrent inputs as well as afferent inputs and the dendritic activity

underwent recurrent gating. (a) As in Fig 2, the trained network segmented two overlapping chunks in the presence of additive recurrent inputs. (b) Normalized average responses of two emergent assemblies during the presentations of chunk 1 and chunk 2. (c) PCA showed that the different chunks were distinguishable by different low-dimensional trajectories, of which the black, orange, and blue portions indicate the periods of random spike input, chunk 1 and chunk 2, respectively.

(PDF)

## Acknowledgments

The authors express their sincere thanks to Thomas Burns for technical assistance.

## Author Contributions

**Conceptualization:** Toshitake Asabuki, Tomoki Fukai.

**Data curation:** Toshitake Asabuki, Tomoki Fukai.

**Formal analysis:** Toshitake Asabuki, Prajakta Kokate.

**Funding acquisition:** Tomoki Fukai.

**Investigation:** Toshitake Asabuki, Tomoki Fukai.

**Methodology:** Toshitake Asabuki.

**Project administration:** Tomoki Fukai.

**Resources:** Tomoki Fukai.

**Software:** Toshitake Asabuki.

**Supervision:** Tomoki Fukai.

**Validation:** Toshitake Asabuki.

**Visualization:** Toshitake Asabuki, Prajakta Kokate.

**Writing – original draft:** Toshitake Asabuki, Tomoki Fukai.

**Writing – review & editing:** Toshitake Asabuki, Tomoki Fukai.

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
