## [Decision Letter · Decision Letter 0]

21 Dec 2021

Dear Dr. Asabuki,

Thank you very much for submitting your manuscript "Neural circuit mechanisms of hierarchical sequence learning tested on large-scale recording data" for consideration at PLOS Computational Biology.

As with all papers reviewed by the journal, your manuscript was reviewed by members of the editorial board and by several independent reviewers. In light of the reviews (below this email), we would like to invite the resubmission of a significantly-revised version that takes into account the reviewers' comments.

The reviewers and I are in agreement that this is a highly interesting and relevant study. To present your results in a more compelling way, the underlying mechanisms must be more clearly exposed and the robustness to parameter settings should be investigated. For the revision, please also make your code available in a link so that reviewers can check it.

We cannot make any decision about publication until we have seen the revised manuscript and your response to the reviewers' comments. Your revised manuscript is also likely to be sent to reviewers for further evaluation.

Sincerely,

Abigail Morrison

Associate Editor

PLOS Computational Biology

Lyle Graham

Deputy Editor

PLOS Computational Biology

Reviewer's Responses to Questions

**Comments to the Authors:**

Reviewer #1: Please find my comments in the PDF attached.

Reviewer #2: In this manuscript, the authors use two-compartments neuron models, with dendritic and somatic compartments, to extract, in an unsupervised manner, representations of the temporal structure of input data. Inspired by previous works from different groups they add recurrent connections between neurons that have the ability to gate signal transmission between the dendritic and somatic compartments. They show that this addition, compared to their previous model that did not use recurrent connections, allows to perform context-dependent segmentation, a feature required for the modeling of hierarchically organized input data.

This work seems definitely interesting, original and important, but I felt it could be improved before publication, in particular to more clearly show the improvement compared to their previous work. The presentation of the work could also be improved, by giving a bit more of explanations on specific points. I detail below my major concerns.

I felt it would be nice to have a sub-section dedicated to the presentation of the model at the beginning of the result section. For now, the presentation of the model is mixed with the first results of the paper, I have found it makes things a bit confusing. And I also felt it would be nice to have a high-level description of the learning procedure, which is not present in the main text if I am correct. In particular, it would be nice to explain how they extended their previous learning algorithm for it to deal with recurrent connections, and say a bit more of the relationship between the formula they derive and the online learning rule they introduce (cf method section). It would also be nice to be able to have, somewhere, a sense of the hyper-parameters that have to be tuned to fit this model.

I have found Fig. 2, 3 and 4 very compelling. Related to the above comment about hyper-parameters, it would be nice to show how these hyper-parameters have been varied before presenting the control result of Fig.S1, that shows that recurrent gating is crucial for context-dependent segmentation.

In Fig.5, the author, if I have understood correctly, want to probe how robust is the learning with respect to variation in the overall temporal scale of the inputs. I would like to see how temporally scaling the input sequence of Fig.3 would affect the down-stream (If I have understood, here the authors introduce another type of input sequence). It would be nice to clearly state in the main text how much warping can be tolerated before neural activity in the model loses its nice context-dependent property. Having such a limit on the degree of acceptable time warping would help better understand the mechanism proposed here as well as its potential limitations. The rationale behind using rank-order correlations could be expanded, for readers not familiar with this measure. Also I did not get what panel e. is about.

In Fig.6, the authors apply their methodology on calcium imaging data. I have been quite confused by this section. In particular, while the previous section states that the mechanism is useful for context-dependent computations, here I did not get whether the behavior of the animals show any form of context-dependencies. Also, it would be nice for readers to understand, how hyper-parameters (e.g. number of neurons in the model) are chosen, and how they impact the result of the analysis.  As for the first data-set, I have understood that the methodology allows to order recorded neurons so as to exhibit nice sequential activations. It seems the procedure they use to get there is quite intricate, aren’t there more simple ways to do so ? Is the gating mechanism crucial here ? As for the second dataset, the presentation of the data is too succinct. For instance to understand what those 500ms correspond to (e.g. is it the whole dataset or a subset chosen as an example ; what is the behavior, besides pupil dilatation, during those 500ms ? ; what is the distinction, in terms of behavior, between the early and late epochs). Here the authors claim that their new method is better than the previous one at exhibiting a form of context dependency (early VS late epoch), but comparing, by eye, figures S4.c and 6.f does not lead to the conclusion of this paragraph. It would be nice to have something more quantitative to substantiate the claim regarding the superiority of the new method.

In Fig.7, the authors want to discuss recent experimental comparisons of « dendritic receptive fields » VS « somatic receptive fields ». The introduction of the corresponding section could be expanded to more clearly state the rationale behind this section. Indeed it can be confusing given the fact that the main message of the paper is centered on the idea of hierarchical sequence, while here there are no notion of hierarchy on the input sequences. Again it would be nice to see whether a model with three dendritic branches, without gating would lead to the same results or not.  It would also be nice to explain how the learning procedure has to be adapted when adding multiple compartments.

 As a side note, I was wondering whether having a model with two branches to fit the hippocampal data for animals on linear track (Fig.6) could give interesting results.

The last part about the role of non-gating recurrent connections was clear and interesting. I again had the same type of comments: would it be possible to describe in the main text how the learning procedure is able to deal with this new connections compared to the gating ones.

Reviewer #3: The authors present a neural network for learning hierarchically organized spike sequences.

This is accomplished by adding a multiplicative gating mechanism, which allows dendritic information to be incorporated in a context-dependent way by the cell bodies. This is a really interesting paper both for its biological insight and for the potential of this network model as a neural activity analysis tool.

I would ask the authors for some clarifications, and possibly to elaborate on the learning capacity of their network.

Fig 1: The circuit diagram could be more informative. In particular there is a gap in detail between the Fig 1a and 1b and the actual equations (1)-(11); for example, where is the “local potential” c? What is its biological meaning? None of the weights or incoming connections are noted on the schematic.

Fig 6: I found the comparison between this and the results without recurrent gating (S4) vague. What do you mean “the cell assembly structures were vague”?

Do you mean they overlap (in cell identities)? From Fig S4a, it does appear that different neurons are active during the forward and backward runs.

pg. 17,18: Define “softmax”

The finding that the network learn assemblies to respond to complex sequences (e.g. ABCD vs. DCBA) is very interesting as is the mechanism for accomplishing this (dendritic activation + gating factors to distinguish between orderings). My question is what is the capacity of the network? i.e. how many such sequences can it learn, and how does this relate to the size of the network?

Can you make a prediction based on the network architecture?

Methods, pg 18: why is there a need for a “standardized potential”; i.e. \\hat{c} and \\hat{V}? Can you explain these quantities.

**Have the authors made all data and (if applicable) computational code underlying the findings in their manuscript fully available?**

Reviewer #1: **No: **The authors note that the code will be make available after acceptance

Reviewer #2: Yes

Reviewer #3: **No: **The authors state code will be available upon acceptance.

PLOS authors have the option to publish the peer review history of their article (what does this mean?). If published, this will include your full peer review and any attached files.

Reviewer #1: No

Reviewer #2: No

Reviewer #3: No
---

## [Decision Letter · Decision Letter 1]

4 Apr 2022

Dear Dr. Asabuki,

Thank you very much for submitting your manuscript "Neural circuit mechanisms of hierarchical sequence learning tested on large-scale recording data" for consideration at PLOS Computational Biology. As with all papers reviewed by the journal, your manuscript was reviewed by members of the editorial board and by several independent reviewers. The reviewers appreciated the attention to an important topic. Based on the reviews, we are likely to accept this manuscript for publication, providing that you modify the manuscript according to the review recommendations, which are largely concerned with providing clarifications.

Sincerely,

Abigail Morrison

Associate Editor

PLOS Computational Biology

Lyle Graham

Deputy Editor

PLOS Computational Biology

[LINK]

Reviewer's Responses to Questions

**Comments to the Authors:**

Reviewer #1: The authors’ additional analysis strengthens the paper and clarifies some of my confusions, but I still have a few outstanding questions:

- It is nice to see the structure of the connectivity matrix in Fig.S4, but it still confuses me how the learning objective, which is to maximize the consistency between the somatic and the gated dendritic potentials, relates to the development of this connectivity structure. I raised this question in my previous review as well, and the authors showed in Fig.S1 that the consistency increases during training. This is nice to confirm, but I think it would really strengthen the paper if the author could further develop the connection between the learning objective and the emergence of sequence-selective cell populations as well as all the other cool properties of the model.

- In figure S4 the authors showed the connectivity matrix W^c., which is helpful to understand the mechanism. Could the authors show the structure of W^v as well? It seems to me that given the right structure in W^c, W^v is not very important for this model to work, since the critical point is the coincidence of the external input and the gating factor c. Is this intuition correct?

Related, could the authors discuss the biological substrates of W^c and W^v? They are both projections from the soma to the neighboring dendrites, so it is not clear to me what might be the difference between their neural implementations.

- I still did not understand the argument that standardizing V and c lets the system avoid a trivial fixed point. I could not find the relevant text in ref.19 except the following sentence "However, as shown previously, the online modifications given in Eqs. (4)–(7) prevent the function \\psi^{som}_i(u_i(t)) from coinciding with \\psi^{dend}(v*_i(t)). This in turn prevents a trivial fixed point w= 0 of Eq. (16)". Could the authors elaborate in more detail the necessity of standardizing V and c, or consequences of not doing so?

Some other minor comments:

- Could the authors specify in more details how the training is done? Are the patterns presented continuously or separated by a long delay (so that network is at baseline when each pattern is presented), or it does not matter (since the neuronal time constant is only 15 ms)?

- For figure 6g, I am afraid I still did not understand the authors' response. Are the number of neurons for the three curves 6532*(1+1), 6532*(1+2/3) and 6532*(1+1/3)? If so, since only 6532 neurons are in the dataset, how did the authors determine the activity of the rest of the neurons?

- In figure 3d and S3c, why do the gating factors for the irrelevant sequence (e.g. orange and green in figure S3c upper panel) rise up during seemingly random periods within a trial?

- Line 347-348: “Non-gating recurrent connections were trained with the same rule as for afferent connections.” It would be more clear to write out the update equations explicitly in the Methods.

- Line 503: “\\tau_s = 5 ms”. Does \\tau_s refer to the \\tau's in equations 12 and 13?

- Line 352-354: “Since only additive recurrent input, but not recurrent gating, can activate postsynaptic cells, the additive input generates a reverberating activity, which may in turn assist the retrieval of learned sequences.” Why can’t recurrent gating activate postsynaptic cells? In general, a network without additive recurrent connections would also be able to generate reverberating activity as well, is that correct?

Reviewer #2: I thank the authors for the work they have produced to address the concerns I raised. I am fully satisfied with their answers, I think this manuscript is ready for publication in PLoS CB.

Reviewer #3: None

**Have the authors made all data and (if applicable) computational code underlying the findings in their manuscript fully available?**

Reviewer #1: Yes

Reviewer #2: Yes

Reviewer #3: Yes

PLOS authors have the option to publish the peer review history of their article (what does this mean?). If published, this will include your full peer review and any attached files.

Reviewer #1: No

Reviewer #2: No

Reviewer #3: No

Figure Files:

Data Requirements:

Reproducibility:

References:

---

## [Decision Letter · Decision Letter 2]

16 May 2022

Dear Dr. Asabuki,

We are pleased to inform you that your manuscript 'Neural circuit mechanisms of hierarchical sequence learning tested on large-scale recording data' has been provisionally accepted for publication in PLOS Computational Biology.

Best regards,

Abigail Morrison

Associate Editor

PLOS Computational Biology

Lyle Graham

Deputy Editor

PLOS Computational Biology

Reviewer's Responses to Questions

**Comments to the Authors:**

Reviewer #1: I thank the authors for their response to my questions and the additional simulation. I have no further questions, and I think the article is suitable for publication on Plos CB.

**Have the authors made all data and (if applicable) computational code underlying the findings in their manuscript fully available?**

Reviewer #1: Yes

PLOS authors have the option to publish the peer review history of their article (what does this mean?). If published, this will include your full peer review and any attached files.

Reviewer #1: No

---

## [Editor Report · Acceptance letter]

9 Jun 2022

PCOMPBIOL-D-21-01729R2 

Neural circuit mechanisms of hierarchical sequence learning tested on large-scale recording data

Dear Dr Asabuki,

I am pleased to inform you that your manuscript has been formally accepted for publication in PLOS Computational Biology. Your manuscript is now with our production department and you will be notified of the publication date in due course.

With kind regards,

Zita Barta
